# Electrical Characteristics and Reliability of Nitrogen-Stuffed Porous Low-*k* SiOCH/Mn_2_O_3−x_N/Cu Integration

**DOI:** 10.3390/molecules24213882

**Published:** 2019-10-28

**Authors:** Yi-Lung Cheng, Yu-Lu Lin, Chih-Yen Lee, Giin-Shan Chen, Jau-Shiung Fang

**Affiliations:** 1Department of Electrical Engineering, National Chi-Nan University, Nan-Tou 54561, Taiwan; willy.lin184@gmail.com (Y.-L.L.); ck7766@gmail.com (C.-Y.L.); 2Department of Materials Science and Engineering, Feng Chia University, Taichung 40724, Taiwan; gschen@fcu.edu.tw; 3Department of Materials Science and Engineering, National Formosa University, Huwei 63201, Taiwan; jsfang@nfu.edu.tw

**Keywords:** porous low-dielectric-constant, barrier, MnO_x_, electrical characteristics, reliability, electric-field acceleration factor, TDDB

## Abstract

In our previous study, a novel barrier processing on a porous low-dielectric constant (low-*k*) film was developed: an ultrathin Mn oxide on a nitrogen-stuffed porous carbon-doped organosilica film (p-SiOCH(N)) as a barrier of the Cu film was fabricated. To form a better barrier Mn_2_O_3−x_N film, additional annealing at 450 °C was implemented. In this study, the electrical characteristics and reliability of this integrated Cu/Mn_2_O_3−x_N/p-SiOCH(N)/Si structure were investigated. The proposed Cu/Mn_2_O_3−x_N/p-SiOCH(N)/Si capacitors exhibited poor dielectric breakdown characteristics in the as-fabricated stage, although, less degradation was found after thermal stress. Moreover, its time-dependence-dielectric-breakdown electric-field acceleration factor slightly increased after thermal stress, leading to a larger dielectric lifetime in a low electric-field as compared to other metal-insulator-silicon (MIS) capacitors. Furthermore, its Cu barrier ability under electrical or thermal stress was improved. As a consequence, the proposed Cu/Mn_2_O_3−x_N/p-SiCOH(N) scheme is promising integrity for back-end-of-line interconnects.

## 1. Introduction

Copper (Cu) conductor and porous low-dielectric-constant (low-*k*) dielectric insulators have been used as back-end-of-line (BEOL) interconnect materials, accelerating the high-speed requirement for advanced microelectronic devices by the reduction of resistance-capacitance (RC) time delay [1,2,3]. However, due to the intrinsically structural and bonding weakness of porous dielectric materials, their integration with Cu is a challenging topic.

Poor adhesion and high diffusivity of Cu are the critical issues, causing integration difficulty of Cu and porous low-*k* dielectric materials [4,5]. Hence, a liner/barrier is interposed between a Cu line and porous dielectric insulator. Currently, a dual-layer of Ta/TaN liner/barrier is used by industry communities [6,7,8]. With the progress of advanced technological nodes, the dimensions of BEOL interconnects are continuously scaling down, and the associated thickness–thinning for Ta/TaN dual-layer results in its failure to maintain effective barrier performance against Cu diffusion. Under such a scenario, a Ta/TaN dual-layer typically deposited by sputtering deposition, occupies a larger portion of the Cu line volume, leading to a huge increase in overall resistivity of lines and vias, as well as a poor step coverage of trenches and vias [9].

Therefore, an alternative to replace the currently used Ta/TaN barrier materials and process for the aggressively scaled-down interconnects is required to overcome the above-mentioned issues, among which self-forming barrier process, chemical vapor deposition (CVD), and atomic layer deposition (ALD) technologies have been proposed [10,11,12]. Moreover, manganese (Mn), ruthenium (Ru), and cobalt (Co), along with their derivatives, are considered as potential alternative barrier materials to substitute the Ta/TaN dual layer [13,14,15].

In light of the aforementioned studies of metal-based barriers, we have recently proposed a novel barrier process for Cu metallization [16] by firstly performing high-pressure nitrogen stuffing of a porous low-*k* SiOCH dielectric. Then, a 2 nm-thick Mn oxide film was deposited in a controlled (oxygen-containing) sputtering gas. Subsequently, an Ar/H_2_ forming-gas treatment was performed on the MnO_x_/porous SiOCH stacked layer to enhance its thermal stability. Such a thermal stabilization treatment results in the transformation of Mn_2_O_3_ and its reaction with stuffed gaseous, ultimately forming an effective ultrathin barrier denoted as Mn_2_O_3−x_N.

This proposed barrier process is an effective barrier against Cu diffusion under thermal stress. However, its impact on the electrical characteristics and reliability of the porous low-*k* SiOCH dielectric remains unclear. Therefore, the purpose of this study is to investigate this topic in detail. The barrier capacities under thermal and electrical stresses for the proposed barrier process are also evaluated.

## 2. Results and Discussion

By recording the plots of relative changes of sheet resistance [(R_T_ − R_0_)/R_0_)] versus annealing temperatures (250–650 °C) for three representative samples, as shown in Figure 1, the effectiveness of Mn_2_O_3_ and Mn_2_O_3−x_N layers as barriers of Cu can be evaluated from comparing the values of temperature corresponding to the onset of an abrupt rise in relative resistance. Notably, R_0_ is the sheet resistance of pristine stacked samples covered by a continuous Cu film, which is thus equivalent to the sheet resistance of the Cu conducting film, while R_T_ is the sheet resistance of the same sample after annealing. Thermal annealing tends to cause Cu to agglomerate and/or penetrate the manganese oxide barrier into the underlying p-SiOCH films [17], leading to failure of the metalized layers. Therefore, the values of (R_T_ − R_0_)/R_0_ reflect the integration of the Cu layers, and thus, the magnitude of threshold temperatures serves to provide the capacity of barrier layers. According to Figure 1, the threshold temperatures for Cu/p-SiOCH (barrier-free) samples and identical samples interposed with Mn_2_O_3_ or Mn_2_O_3−x_N barriers are 450 °C, 550 °C, >550 °C, respectively. Notably, the limit of temperature for BEOL interconnect processing is around 400 °C. Therefore, the slight increase in (R_T_ − R_0_)/R_0_ at the threshold temperature (400 °C) for Cu/p-SiOCH indicates the necessity of a barrier. Comparing the magnitudes of the threshold temperature in Figure 1 also suggests that Mn_2_O_3−x_N outperforms Mn_2_O_3_; it exhibits an outstanding barrier capacity with a significantly higher threshold temperature (650 °C versus 550 °C), even with a thickness of only 2 nm.

Table 1 lists the results of the tape testing of the as-deposited and annealed (450 °C for 2 h) samples, including Cu/p-SiOCH/Si, Cu/Mn_2_O_3_/p-SiOCH(N)/Si, and Cu/Mn2O_3−x_N/p-SiOCH(N)/Si. The size of the tested sample films was 6 × 6 square regions. For the pristine and annealed (barrier-free) Cu/p-SiOCH/Si samples, no square or only a few square regions were left after tape testing, indicating the delamination of Cu films due to poor adhesion. For the pristine Cu/Mn_2_O_3_/p-SiOCH(N)/Si samples, about 20% of the square regions were delaminated. After annealing, the delamination percentage was slightly increased to approximately 35%, indicating that thermal annealing degrades the interfacial adhesion. On the contrary, all the 36 square regions in the Cu/Mn_2_O_3−x_N/p-SiOCH(N)/Si samples remained intact before and after annealing, indicating that, apart from acting as a barrier, the 2 nm-thick Mn_2_O_3−x_N layer serves as an adhesion promoter, thus markedly increasing the adhesion of Cu films. The significant enhancement of interfacial adhesion by Mn_2_O_3−x_N is attributed to a chemical interaction between contacted layers and the resulting nitrogen-related bonding between the Cu metal electrode and the p-SiOCH substrate. The obtained result is similar to the reported finding by Selvaraju et al., indicating that a noticeable improvement in adhesion of Cu to the underlying SiO_2_ was observed by the use of Mn-nitride with a nitrogen content of just 5% [18].

It is important for the underlying p-SiOCH layers to keep a low-*k* feature after completing the fabrication of the Cu/Mn_2_O_3−x_N/p-SiOCH(N)/Si structure, involving nitrogen stuffing, Mn_2_O_3_ deposition, stabilization annealing [converting Mn_2_O_3_ to Mn_2_O_3−x_N], and Cu deposition. Therefore, the variations of the accumulation capacitances (*C*) and corresponding *k* values of the p-SiOCH layers after the aforementioned treatment sequence were determined using *C-V* measurements, as plotted in Figure 2. For the pristine p-SiOCH film, the accumulation capacitance per area was 2.266 × 10^−4^ F/m^2^, corresponding to a *k* value of 2.560. After a nitrogen-stuffing treatment at 450 °C/4 atmosphere for 5 min, the accumulation capacitance slightly increased to 2.317 × 10^−4^ F/m^2^, leading to an increase of *k* value to 2.618. Notably, the high-pressure nitrogen treatment results in the stuffing of the pores within the p-SiOCH film [denoted as p-SiOCH(N)] with physisorbed nitrogen atoms (or molecules) [16], thus rendering only a 2.26% increase in *k* value. After the sputter deposition of a 2 nm-thick Mn_2_O_3_ barrier, a large increase in accumulation capacitance of the p-SiOCH(N) film was measured, associated with a marked increase of *k* value to 2.914 (11.30%). This increase is attributed to two mechanisms: one is plasma-induced damage on the p-SiOCH(N) film during the deposition of the Mn_2_O_3_ barrier [19]; another is the relative high *k* value of 5.10 of the coated Mn_2_O_3_ [20,21]. After annealing with Ar/H_2_ forming gas at 450 °C for 1 h, the Mn_2_O_3_ film became more stabilized. Moreover, the nitrogen stuffed into the pores in the p-SiOCH(N) film was released and reacted with the Mn_2_O_3_ film to form the Mn_2_O_3−x_N layer. This transformation reduced the capacitance, resulting in a reduction of the *k* value to 2.845. The capacitance and *k* value slightly decreased as the Cu gate was formed. Further annealing (450 °C/2 h) of the Cu/Mn_2_O_3−x_N/p-SiOCH(N)/Si capacitor in an N_2_ environment made the *k* value slightly decrease to 2.757. It is worth mentioning that the *k* value of Cu/p-SiOCH/Si (without a barrier) capacitors decreased upon annealing with a decreasing magnitude of 0.214, which could be caused by Cu diffusion [22]. As a result, a lower decrease in the *k* value for Cu/Mn_2_O_3−x_N/p-SiOCH(N)/Si capacitor upon annealing is indicative of greater Cu barrier capacity of the Mn_2_O_3−x_N layer.

Figure 3 plots the *I-E* curves of various Cu-gate capacitors before and after annealing. The negative-polarity voltage was applied and transformed to the electric field (*E*) by dividing it by the thickness of the p-SiOCH films. For all Cu-gate capacitors, the leakage currents increased with the electric fields, typically followed by a plateau, and then suddenly jumped. The applied field on the threshold of this jump is defined as a breakdown field of the p-SiOCH films. Prior to annealing and before the breakdown of the p-SiOCH films, leakage currents of the capacitor samples with and without a Mn_2_O_3_ barrier layer were comparable, while those of the Cu/Mn_2_O_3−x_N/p-SiOCH(N)/Si capacitors were substantially reduced by 1–2 orders. This finding indicates that the Mn_2_O_3−x_N layer has a better barrier performance than the Mn_2_O_3_ layer, thus significantly improving the electrical properties of p-SiOCH films. Furthermore, the conduction of leakage currents in dielectric materials is mainly through bulk transportation and surface migration of Cu atoms and ions. Therefore, leakage currents are ascribed to various factors, such as chemical structures of dielectrics, incorporated impurities, and metal-dielectric interfaces [23]. Since the Cu-gate capacitors studied here had the identical p-SiOCH film, the decrease in leakage current for the Cu-gate capacitor with a Mn_2_O_3−x_N barrier layer is attributed to the improvement of metal-dielectric interfaces, thereby retarding surface migration of Cu atoms. After annealing, the leakage currents of all Cu-gate capacitors increased over the whole range of applied fields. The increase was most pronounced for the barrier-free samples, while the capacitor samples with a Mn_2_O_3−x_N barrier layer exhibited the least increase in leakage currents. This implies that the conduction via a metal-dielectric interface is the dominant mechanism for the leakage current.

Figure 4 compares the breakdown fields of p-SiOCH films in various Cu-gate capacitors before and after annealing. Ten samples were measured for each condition. Before annealing, the magnitudes of breakdown fields followed the order of: Mn_2_O_3_ capped sample (8.74 MV/cm) > barrier-free sample (8.46 MV/cm) > Mn_2_O_3−x_N capped sample (7.48 MV/cm). This trend is different from the result of leakage currents, suggesting that the leakage current is not the necessary condition to trigger the breakdown of a dielectric film, which yields a reduced value of the breakdown field as its molecular bonds are broken up. For the p-SiOCH film in the Cu-gate capacitor with a Mn_2_O_3−x_N barrier layer, additional annealing at 450 °C for 1 h was performed on it before the Cu gate deposition. The broken bonds were expected, thereby reducing the breakdown strength. Additionally, the barrier-free sample had a lower breakdown field than the Mn_2_O_3_ capped sample due to the migration of Cu atoms/ions into the p-SiOCH film during the Cu-gate deposition. This suggests that a barrier is required for Cu/p-SiOCH integrity. After annealing, the breakdown fields of p-SiOCH films in all the capacitor samples were reduced. The largest magnitude of decreasing occurred in the barrier-free capacitor sample because thermal stress induces bond breakage of p-SiOCH film and migration of Cu. With a Mn_2_O_3_ or Mn_2_O_3−x_N barrier, Cu migration was inhibited. As a result, the reduction in the breakdown field was not as large as that obtained from the barrier-free sample. In the case of the Mn_2_O_3−x_N barrier, its magnitude of decreasing was the lowest, but the breakdown field variation was the smallest, suggesting that the Mn_2_O_3−x_N layer is the best barrier against Cu thermal diffusion.

After thermal stress, *C-V* characteristics were measured again. Normalized *C-V* curves for the barrier-free and barrier capped capacitor samples before and after thermal stress (450 °C/2 h) were compared, and the shifts of their flat-band voltages (V_fb_) were determined, as presented in Figure 5. The data presented here are the average values from three measurements for different capacitor samples. After thermal stress, the barrier-free capacitor sample exhibited a larger V_fb_ shift with a negative direction. Additionally, its accumulation capacitance obviously decreased. These features represent the diffusion of Cu ions into the p-SiOCH film during thermal stressing [24]. For the Mn_2_O_3_ capped capacitor sample, V_fb_ still shifted toward negative voltage, but the shifting magnitude was obviously reduced. This result indicates that the diffusion of Cu ions into the p-SiOCH film is blocked by the Mn_2_O_3_ barrier, although, its blocking efficiency cannot reach 100%. On the other hand, for the Mn_2_O_3−x_N capped capacitor sample, a small V_fb_ shift was seen, but the direction was toward positive voltage. This positive shift of V_fb_ indicates the presence of negative charges, but not Cu diffusion into the p-SiOCH film, which in turn suggests that the retardation of Cu diffusion under thermal stress is ascribed to the Mn_2_O_3−x_N barrier and nitrogen-stuffed p-SiOCH film.

In summary, Cu/Mn2O_3−x_N/p-SiOCH(N) integration has been demonstrated to possess the best capacity against Cu thermal diffusion. However, under an electrical stress, Cu ions would drift into the p-SiOCH(N) (porous) dielectric film. Indeed, such phenomena are more pronounced for a porous film [25,26]. As the Cu-gate is stressed with a positive bias, Cu ions are generated and then drift into a dielectric film under the external electric field. After electrical stress, *I-V* characteristics with a sweep at a negative voltage were measured. The measured current is the response of both injection carriers and ionic currents.

Figure 6a plots the *I-V* curves of Mn_2_O_3−x_N capped capacitor samples before and after electrical stress (2.0 MV/cm) for various times. After electrical stress, the leakage current increased. Moreover, as the stressing time reached a certain value, a hump appeared at the voltage range from 0 to −20 V. The increase in the leakage currents is ascribed to the stress-induced trapping of within the p-SiOCH films. The appearing hump is mainly caused by ionic currents due to the migration of Cu ions. Under positive-polarity electrical stresses, Cu ions were generated, subsequently drifting into the p-SiOCH films following the assistance of the external electric fields. Then, during the application of negative-polarity voltages, the drifted Cu ions reversed back to the metal gate. Therefore, the presence of such a hump can be an indicator of Cu barrier capacity under an electrical stress. In this study, “Cu diffusion time” is defined as the stressing time of the positive-polarity stress required to produce this hump. A longer Cu diffusion time means a better Cu barrier capacity. Figure 6b compares the Cu diffusion times of the three capacitor sample types as a function of the electric field. The Cu diffusion times decrease with increases in the electric field for all the samples, indicating that a high electric field generates more Cu ions with faster drift rates. As expected, Cu-gate capacitors without a Mn_2_O_3_ barrier layer have the lowest Cu diffusion time. Cu-gate capacitors with Mn_2_O_3_ and Mn_2_O_3−x_N barrier layers have a longer and comparable Cu diffusion time, indicating that both barrier layers (Mn_2_O_3_ and Mn_2_O_3−x_N) have a similar capacity in retarding Cu migration against electrical stress.

The long-term reliability of p-SiOCH films in the Cu-gate MIS capacitors with various barriers was evaluated by comparing the results of time-dependent-dielectric-breakdown (TDDB) tests with those of barrier-free samples. During a TDDB test (at 25 °C), the samples were stressed by a constant electric field, and the leakage current was monitored until dielectric breakdown. The time-to-fail (TTF) was recorded as a leap in leakage current by at least three orders of magnitude. Herein, at least three values of electric fields were used for stressing, and twelve MIS capacitors were measured for each condition, and the data of median TTFs for various samples (before and after annealing) are presented in Figure 7. Similar to the results of the breakdown fields measured from *I-V* measurements (Figure 4), the measured TTFs display the same order. That is, the p-SiOCH film in the Mn_2_O_3_ capped MIS capacitor exhibited the largest TTFs as compared to those in other MIS capacitors. On the contrary, the Cu/Mn_2_O_3−x_N/p-SiOCH(N)/Si capacitor had the shortest TTFs. After annealing, the TTFs of the p-SiOCH films in all the Cu-gate MIS capacitors were reduced. Similarly, the p-SiOCH film in the Mn_2_O_3_ capped capacitor had the largest TTFs. On the other hand, thermal annealing caused the lowest reduction in TTFs for the Cu/Mn_2_O_3−x_N/p-SiOCH(N)/Si capacitor. As a result, its TTFs were slightly larger than those obtained from the barrier-free samples. The result reveals that a barrier is required to ensure reliability for Cu and p-SiOCH integrity.

Annealing results in a reduction of TTFs; it also causes a change of slopes in the plots of TTF versus the stressing filed, representing that the annealing causes an alteration of electric-field acceleration factors. In this study, the E model [TTF = A exp (−γE)], which is the key to bond-breakage mechanism, was used to describe the dielectric breakdown behavior [27,28]. In the equation, γ is the electric-field acceleration factor, representing the extent to which TTF (derived from TDDB tests) is affected by electric fields. A larger γ value means that TTF has a stronger dependence on electric fields. Moreover, it provides a longer lifetime in lower operation fields. The γ values extracted from the plots in Figure 7 for the three different capacitor types before and after annealing are plotted in Figure 8. The barrier-free capacitor sample before annealing had a lower γ value (2.42), which was significantly reduced to 1.62 (the lowest). The decrease of γ value is believed to be caused by the drifting of Cu ions [24]. Both capacitor samples with Mn2O3 and Mn2O3-xN barriers had the similar γ value of 3.12–3.13. Upon annealing, a decrease in γ value to 2.86 was found in the Mn2O3 capped capacitor, whereas a slight increase to 3.23 occurred for the Mn2O3-xN capped capacitor, indicating better Cu barrier efficiency of the Mn2O3-xN layer. The large γ value for the Cu/Mn2O3-xN/p-SiOCH(N)/Si capacitor has a dramatic benefit of promoting a dielectric lifetime in a low operation field. As a result, the lifetimes of the Cu-gate MIS capacitor with the Mn2O3-xN barrier layer surpass that obtained from the Cu MIS capacitors with the MnOx barrier layer when the operation field is lower than 4.06 MV/cm.

## 3. Experiments

The low dielectric-constant (*k*) material used in this study was porous SiOCH (abbreviated hereafter as p-SiOCH), which was deposited on a *p*-type (100) Si wafer by an Applied Materials plasma-enhanced chemical vapor deposition system using diethoxymethylsilane (DEMS), oxygen (O_2_), and alpha-terpinene (ATRP) porogen as precursors [29]. The thickness and *k* value of p-SiOCH dielectric films were ~100 nm and 2.560, respectively.

First, a p-SiOCH sample was placed into a vacuum furnace evacuated by a scroll pump to background pressure of approximately 1 Pa. The p-SiOCH film was then stuffed at 450 °C for 5 min under 4 atm of surrounding nitrogen, hereafter denoted as p-SiOCH(N). Subsequently, a 2 nm-thick layer of Mn_2_O_3_ (transmission electron microscopy; Hitachi HD-2300A, Hitachi–Science & Technology, Tokyo, Japanese) was deposited onto p-SiOCH(N) through a shadow mask by sputtering a Mn target under a controlled Ar/O_2_ oxidative atmosphere. The resultant sample, denoted as Mn_2_O_3_/p-SiOCH(N)/Si, was subjected to stabilization annealing at 450 °C in an Ar/H_2_ (5%) forming gas for 60 min, converting the Mn_2_O_3_ to Mn_2_O_3−x_N [16]. Finally, a 50 nm-thick Cu film (alpha-step 200 profilometer, KLA/Tencor, Austin, TX, USA) was thermally evaporated onto the stacked sample also through a shadow mask. The area of the formation Cu electrode was 9.0 × 10^−4^ cm^2^. Based on this process flow, metal-insulator-silicon (MIS) capacitors with a Cu/Mn_2_O_3−x_N/p-SiOCH(N)/Si stacked structure were fabricated for electrical characteristics and reliability measurements. Other MIS capacitors with Cu/p-SiOCH/Si and Cu/Mn_2_O_3_/p-SiOCH/Si stacked structures were also fabricated for a reference.

*C-V* plots of dielectric films were measured by a semiconductor parameter analyzer (HP4280A, Agilent technologies, Santa Clara, CA, USA), using applied voltages in the range from −40 V to 40 V and −40 V and an operation frequency of 1 MHz. The *k* value of each dielectric film was derived from the measured accumulation capacitance value (*C*) of the associated *C-V* plots, along with the values of the Cu electrode area (A) and thickness (*d*) of the dielectric film, using the equation: *C* = *ε*_o_*k*A/*d*, where *ε*_o_ is absolute capacitivity in vacuum ( 8.85 × 10^−12^ F/m). Furthermore, leakage currents of a dielectric film were measured versus applied voltages by a ramped-voltage-stress (RVS) method with a ramping rate of 0.1 V/s, from which a breakdown field was derived. Dielectric breakdown time was detected from time-dependent-dielectric-breakdown (TDDB) measurements by applying a constant electric voltage (field). RVS and TDDB tests were performed by an electrometer (Keithley, 6517A, Austin, TX, USA). All measurements were performed at room temperature (25 °C). The electrical aspect of the diffusion barrier properties for the Cu/Mn_2_O_3−x_N/p-SiOCH(N)/Si structure was evaluated by a thermal stress at 450 °C for 2 h and electrical stress at 1.5–3.0 MV/cm.

## 4. Conclusions

The electrical characteristics and reliability of the integrated Cu/Mn_2_O_3−x_N/p-SiOCH(N)/Si structure have been investigated in this study. The Cu/Mn2O3-xN/p-SiOCH(N)/Si structure had better adhesion and Cu barrier capacity. However, it exhibited poor dielectric breakdown characteristics in the as-fabricated stage. After thermal stress, less degradation was observed. Moreover, its TDDB electric-field acceleration factor slightly increased, leading to a larger dielectric lifetime in a low operating field as compared to other MIS capacitors. Consequently, the proposed Cu/Mn2O3-xN/p-SiOCH(N) is a promising integration processing for Cu/porous low-k interconnects.

## Figures and Tables

**Figure 1 molecules-24-03882-f001:**
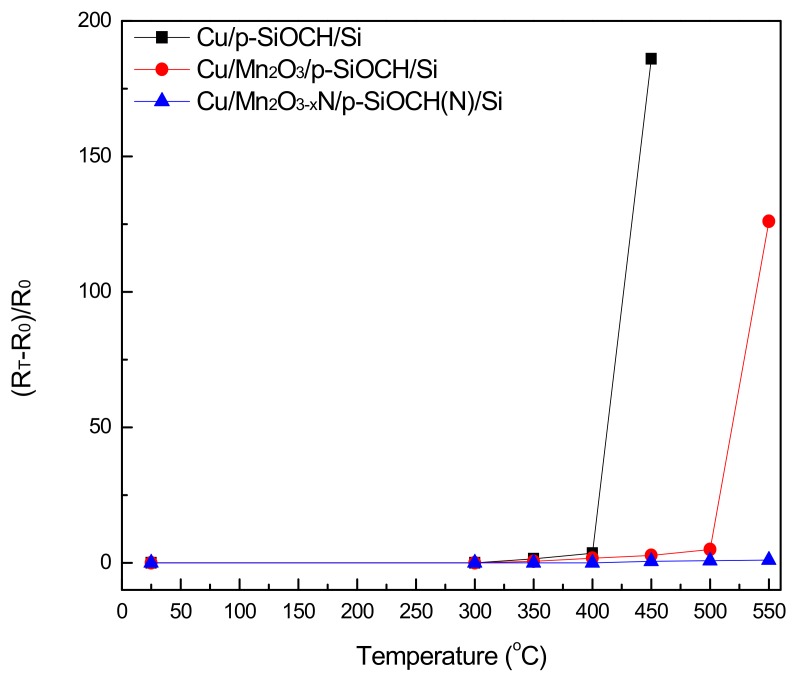
Variation of relative electrical resistivity as a function of annealing temperature for various Cu-gate metal-insulator-silicon (MIS) capacitors.

**Figure 2 molecules-24-03882-f002:**
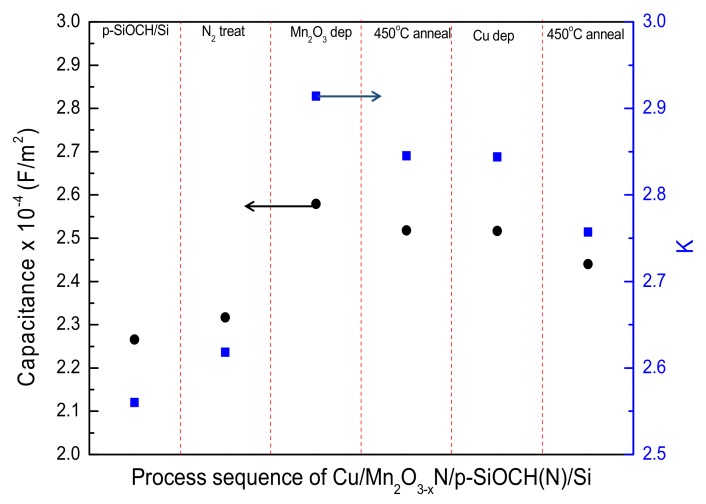
Variations in capacitance and dielectric constant during Cu/Mn_2_O_3−x_N/p-SiOCH(N)/Si integration.

**Figure 3 molecules-24-03882-f003:**
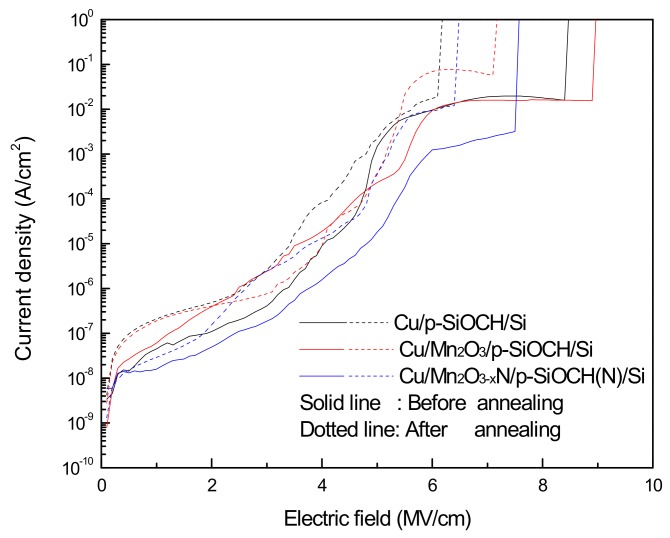
*J-E* plots of various Cu-gate MIS capacitors before and after annealing at 450 °C for 2 h.

**Figure 4 molecules-24-03882-f004:**
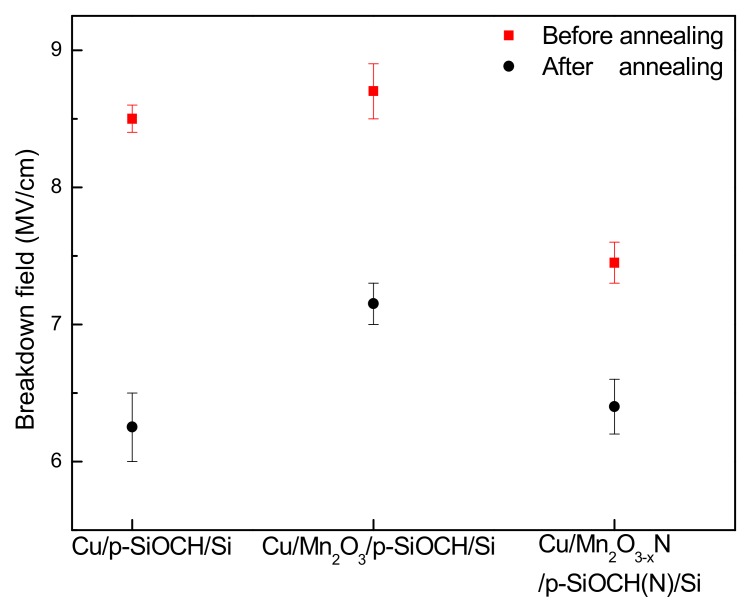
Comparison of breakdown fields of various Cu-gate MIS capacitors before and after annealing.

**Figure 5 molecules-24-03882-f005:**
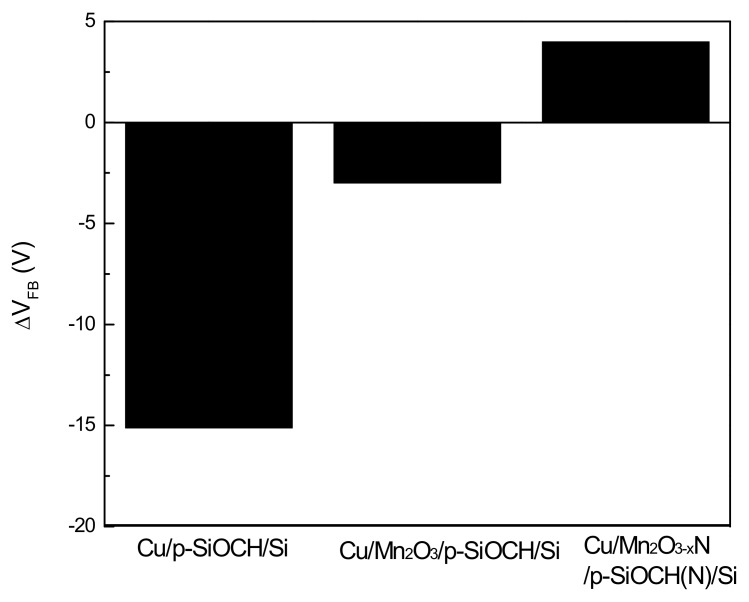
Shift of the flatband voltage (V_FB_) of various Cu-gate MIS capacitors after being subjected to thermal stress.

**Figure 6 molecules-24-03882-f006:**
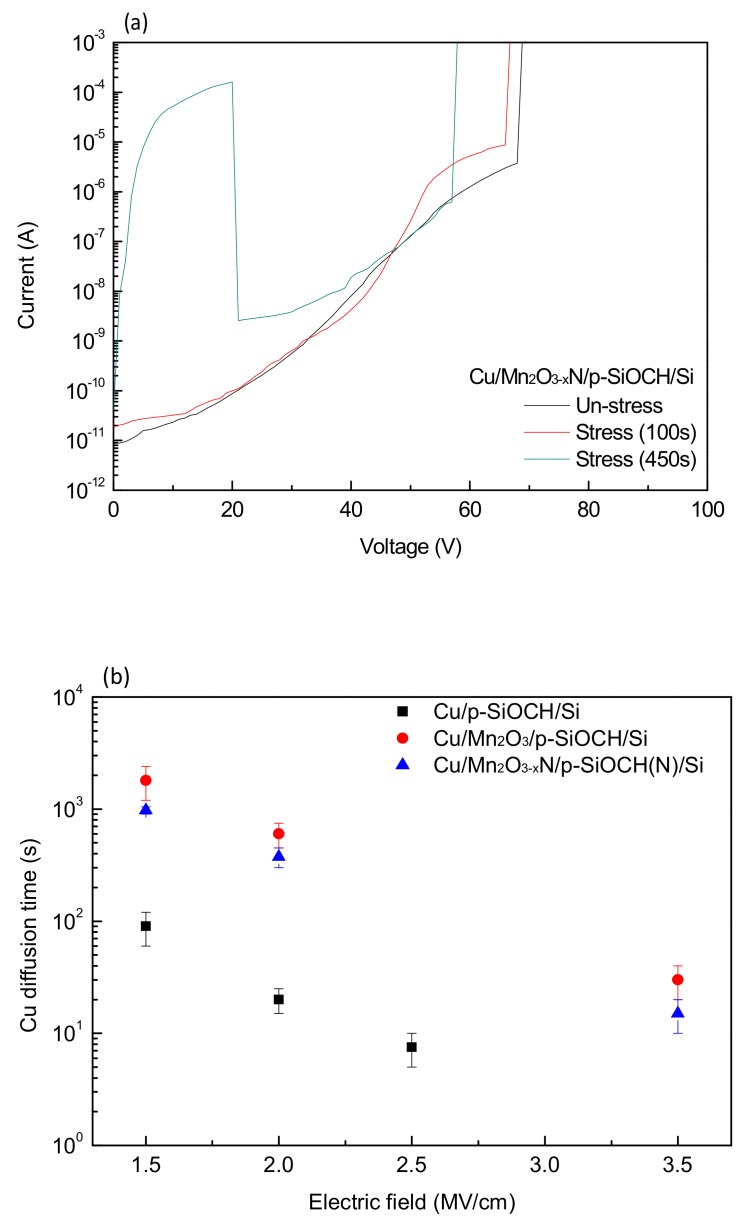
(**a**) *I-V* curves of Cu-gate an MIS capacitor with a Mn_2_O_3−x_N barrier before and after electrical stress (2.0 MV/cm) for various times; (**b**) Cu diffusion time of various Cu-gate MIS capacitors under electrical stress as a function of the electric-field.

**Figure 7 molecules-24-03882-f007:**
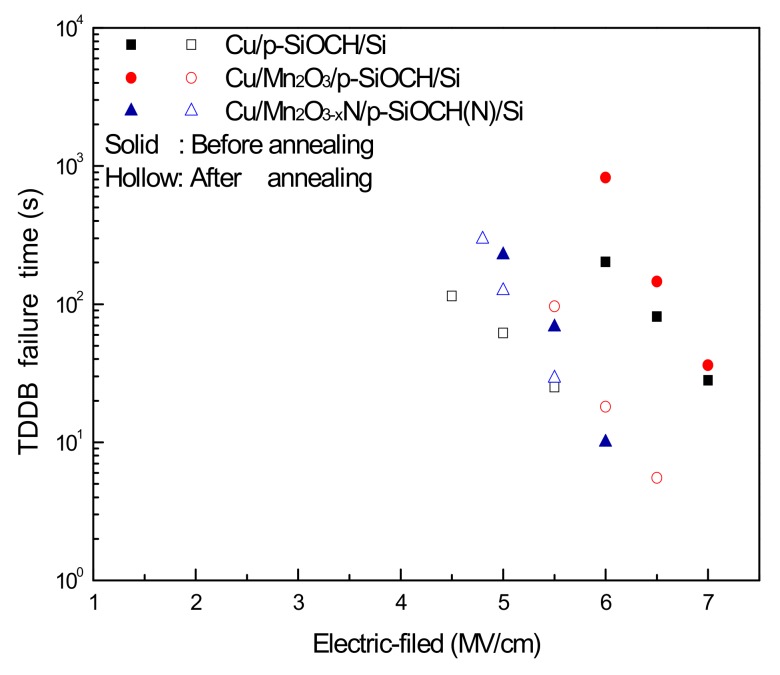
The time-to-fails (TTFs) of p-SiOCH film in various Cu-gate MIS capacitors as a function of the applied electric-field before and after annealing.

**Figure 8 molecules-24-03882-f008:**
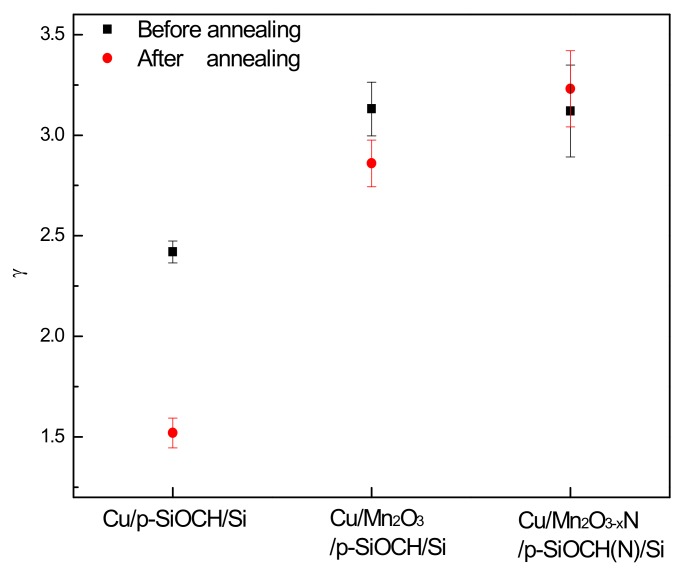
Electric-field acceleration factor of various Cu-gate MIS capacitors before and after annealing.

**Table 1 molecules-24-03882-t001:** Tape test results of various MIS capacitors before and after annealing.

	Cu/p-SiOCH/Si	Cu/Mn_2_O_3_/p-SiOCH/Si	Cu/Mn_2_O_3−x_N/p-SiOCH/(N)Si
As dep.	Fail	~20% Fail	Pass
450 °C/2 h	N/A	~35% Fail	Pass

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
