# Peer review of "Electrical Characteristics and Reliability of Nitrogen-Stuffed Porous Low-k SiOCH/Mn2O3−xN/Cu Integration"

_molecules, 2019, doi:10.3390/molecules24213882_

Round 1

Reviewer 1 Report

The manuscript entitled “Electrical Characteristics and Reliability of Nitrogen-Stuffed Porous Low-k SiOCH/ Mn2O3-xN/Cu Integration” study on the the electrical characteristics and reliability of this integrated Cu/Mn2O3-xN/p-SiOCH(N)/Si structure. Cu/Mn2O3-xN/p-SiOCH(N)/Si capacitors exhibited poor dielectric breakdown characteristics in the as-fabricted stage, although, less degradation was found after thermal stress. The barrier property under electrical or thermal stress was improved. The manuscript is well-presented with scientific sound and logic. However, there still exist few errors in the manuscript, therefore I recommend the authors to do a revision the manuscript before reconsider for publication in Molecules. Detailed comments as following:

- Please provide the surface and edge morphology of Cu/Mn2O3-xN/p-SiOCH(N)/Si structure, it could be optical or SEM or TEM images.

- How do you prove the successful formation of Mn2O3-xN after nitrogen stuffing? Need to provide the evidences.

- What is the thickness of Cu films, how do you measure it?

- Line 121 – 122, the author claimed that “2 nm-thick Mn2O3-xN layer”, please provide the method to measure this thickness.

- Conclusion is too brief, please summarize more details resulted from this work.

- In the line 49 and 72, the author cited the wrong reference, it should be “reference 17”instead of 16.

- Several English errors should be corrected, for example:

+ In the abstract, verb should be used in the past tense.

+ Line 22, “consequent” should be “consequence”

Author Response

The manuscript entitled “Electrical Characteristics and Reliability of Nitrogen-Stuffed Porous Low-k SiOCH/ Mn2O3-xN/Cu Integration” study on the the electrical characteristics and reliability of this integrated Cu/Mn2O3-xN/p-SiOCH(N)/Si structure. Cu/Mn2O3-xN/p-SiOCH(N)/Si capacitors exhibited poor dielectric breakdown characteristics in the as-fabricted stage, although, less degradation was found after thermal stress. The barrier property under electrical or thermal stress was improved. The manuscript is well-presented with scientific sound and logic. However, there still exist few errors in the manuscript, therefore I recommend the authors to do a revision the manuscript before reconsider for publication in Molecules. Detailed comments as following:

- Please provide the surface and edge morphology of Cu/Mn2O3-xN/p-SiOCH(N)/Si structure, it could be optical or SEM or TEM images.

Ans: This study focuses on the electrical Characteristics and Reliability. Material analysis can be found in ref. [16].

- How do you prove the successful formation of Mn2O3-xN after nitrogen stuffing? Need to provide the evidences.

Ans: Material analysis can be found in ref. [16]. From XPS analysis, Nitrogen signal can be detected.

- What is the thickness of Cu films, how do you measure it?

Ans: We added “a 50-nm-thick Cu film (alpha-step 200 profilometer) was thermally evaporated (alpha-step 200 profilometer)”in the experimental part.

- Line 121 – 122, the author claimed that “2 nm-thick Mn2O3-xN layer”, please provide the method to measure this thickness.

Ans: We provided the information in in the experimental part. “a 2-nm-thick layer of Mn2O3 (transmission electron microscopy; Hitachi HD-2300A) was deposited onto…..”

- Conclusion is too brief, please summarize more details resulted from this work.

Ans: We had revised conclusion in the revised manuscript.

- In the line 49 and 72, the author cited the wrong reference, it should be “reference 17”instead of 16.

Ans: Ref. had been checked and revised.

- Several English errors should be corrected, for example:

+ In the abstract, verb should be used in the past tense.

+ Line 22, “consequent” should be “consequence”

Ans: English errors had been corrected.

Reviewer 2 Report

In this article, the authors proceeded some electrical characteristics and reliability of Cu/Mn2O3-xN/p-SiOCH(N)/Si, which was expected to be applied as back-end-of-line interconnects. However, much more experiments should be supplemented before it can be considered to whether be received by this journal or not. The main problems are pointed out as follows:

1.There were no experiments for the characterization of the materials, and therefore, the structure of the materials was ambiguous. The corresponding information cannot be found in the previous paper which was cited by the authors, ether.

2.Only the electrical characteristics were demonstrated in spite of the fact that as an excellent low-k interconnects, other properties, such as high thermo conductivity, were equally important, which can help the electronical devices to get rid of the heat dissipation difficulties. The current characteristics were insufficient without other tests to ensure the practical application as a back-end-of-line interconnects. The tests in this article were too monotonous to support a fully developed paper.

3.Some language mistakes were made, among which some are stated. “was released and react” should be “was released and reacted”(line 145); “in is worth mentioning” should be “it is worth mentioning”(line 149); there were no predicate verb in the sentence in line 190 and line 191. As a result, the article should be carefully checked by the authors to avoid similar mistakes.

Overall, the paper should be further developed and well organized, after which it can be submitted to this journal and be considered again.

Author Response

In this article, the authors proceeded some electrical characteristics and reliability of Cu/Mn2O3-xN/p-SiOCH(N)/Si, which was expected to be applied as back-end-of-line interconnects. However, much more experiments should be supplemented before it can be considered to whether be received by this journal or not. The main problems are pointed out as follows:

There were no experiments for the characterization of the materials, and therefore, the structure of the materials was ambiguous. The corresponding information cannot be found in the previous paper which was cited by the authors, ether.

Ans: This study focuses on the electrical Characteristics and Reliability. Material analysis can be found in ref. [16].

2.Only the electrical characteristics were demonstrated in spite of the fact that as an excellent low-k interconnects, other properties, such as high thermo conductivity, were equally important, which can help the electronical devices to get rid of the heat dissipation difficulties. The current characteristics were insufficient without other tests to ensure the practical application as a back-end-of-line interconnects. The tests in this article were too monotonous to support a fully developed paper.

Ans: This study did not measure thermo conductivity. Review’s comment is very constructive. We will investigate this topic in the future work.

Some language mistakes were made, among which some are stated. “was released and react” should be “was released and reacted”(line 145); “in is worth mentioning” should be “it is worth mentioning”(line 149); there were no predicate verb in the sentence in line 190 and line 191. As a result, the article should be carefully checked by the authors to avoid similar mistakes.

Ans: English errors had been corrected.

Reviewer 3 Report

The paper presents the electrical characteristics and reliability of nitrogen-stuffed porous low-k SiOCH/Mn2O3-xN/Cu integration. According to the reviewer’s opinion, the paper is well-structured and clear. The topic is interesting and falls within the aim of the journal. In addition, the results are well-presented and could be helpful to further develop the same topic. Therefore, the paper can be accepted for publication in the current form.

Author Response

The paper presents the electrical characteristics and reliability of nitrogen-stuffed porous low-k SiOCH/Mn2O3-xN/Cu integration. According to the reviewer’s opinion, the paper is well-structured and clear. The topic is interesting and falls within the aim of the journal. In addition, the results are well-presented and could be helpful to further develop the same topic. Therefore, the paper can be accepted for publication in the current form.

Ans: Thanks for reviewer’s positive comment.

Round 2

Reviewer 2 Report

The article can be accepted and published in the present form.